evolution/molecular biology/taxonomy and systematics

mitochondrial genome, gene order, origin of replication, inversion, Cymothoida, GC skew

**Author for correspondence:**
Gui-Tang Wang
e-mail: gtwang@ihb.ac.cn

# Architectural instability, inverted skews and mitochondrial phylogenomics of Isopoda: outgroup choice affects the long-branch attraction artefacts

Hong Zou[1], Ivan Jakovlić[2], Dong Zhang[1,3], Cong-Jie Hua[1,4], Rong Chen[2], Wen-Xiang Li[1,3], Ming Li[1,3] and Gui-Tang Wang[1,3]

[1]Key Laboratory of Aquaculture Disease Control, Ministry of Agriculture, and State Key Laboratory of Freshwater Ecology and Biotechnology, Institute of Hydrobiology, Chinese Academy of Sciences, Wuhan 430072, People's Republic of China
[2]Bio-Transduction Lab, Wuhan 430075, People's Republic of China
[3]University of Chinese Academy of Sciences, Beijing 100049, People's Republic of China
[4]Department of Pathogenic Biology, School of Medicine, Jianghan University, Wuhan 430056, People's Republic of China

The majority strand of mitochondrial genomes of crustaceans usually exhibits negative GC skews. Most isopods exhibit an inversed strand asymmetry, believed to be a consequence of an inversion of the replication origin (ROI). Recently, we proposed that an additional ROI event in the common ancestor of Cymothoidae and Corallanidae families resulted in a double-inverted skew (negative GC), and that taxa with homoplastic skews cluster together in phylogenetic analyses (long-branch attraction, LBA). Herein, we further explore these hypotheses, for which we sequenced the mitogenome of *Asotana magnifica* (Cymothoidae), and tested whether our conclusions were biased by poor taxon sampling and inclusion of outgroups. (1) The new mitogenome also exhibits a double-inverted skew, which supports the hypothesis of an additional ROI event in the common ancestor of Cymothoidae and Corallanidae families. (2) It exhibits a unique gene order, which corroborates that isopods possess exceptionally destabilized mitogenomic architecture. (3) Improved taxonomic sampling failed to resolve skew-driven phylogenetic artefacts. (4) The use of a

single outgroup exacerbated the LBA, whereas both the use of a large number of outgroups and complete exclusion of outgroups ameliorated it.

# 1. Introduction

The phylogeny of Isopoda (class Malacostraca), a highly speciose order of crustaceans, remains debated, with different datasets (mitochondrial genes, mitochondrial genomes, nuclear genes, combined mitonuclear data) often producing starkly contradictory phylogenetic hypotheses [1–3]. Along with several 'rogue' taxa, the status of the Cymothoida suborder is particularly contentious: it has been resolved as deeply paraphyletic by some studies and datasets, with Cymothoidae and Corallanidae species clustering at the base of the entire isopod clade, and monophyletic and highly derived within the Isopoda by other studies/datasets [2–11]. We have recently shown that some of these contradictory hypotheses can be attributed to asymmetrical skews in the base composition of mitochondrial genomes, which interfere with phylogenetic reconstruction by producing long-branch attraction (LBA) artefacts [1]. Organellar genomes often exhibit a phenomenon known as strand asymmetry, or strand compositional bias, where positive AT skew values indicate more A than T on the strand, positive GC skews indicate more G than C, and vice versa [12,13]. Whereas crustacean taxa usually exhibit negative overall GC skews and positive AT skews for genes located on the majority strand (the strand encoding a majority of genes, also referred to as the plus strand) [13–15], this strand bias is inversed in isopod mitogenomes, which generally exhibit negative AT skews and positive GC skews on the majority strand [5,10,16]. This is believed to be a consequence of an inversion of the replication origin (ROI) located in the control region (CR), where the changed replication order of two mitochondrial DNA strands consequently resulted in an inversed strand asymmetry [5,13,15–17]. Until very recently, the only known exceptions among the available isopod mitogenomes were two species from the suborder Asellota (*Asellus aquaticus* and *Janira maculosa*), which possess standard crustacean skews [5]. We found that three species belonging to two parasitic Cymothoida (suborder) families, Cymothoidae (*Cymothoa indica*, and *Ichthyoxenos japonensis*) and Corallanidae (*Tachaea chinensis*), also exhibit standard crustacean skews, i.e. inverted skews in relation to other isopods, here referred to as double-inverted skews (D-I skews) [1]. These species clustered with the Asellota at the base of the isopod clade, which initially made us hypothesize that the ROI in isopods occurred after the split of these two clades (Cymothoidae + Corallanidae and Asellota). However, this hypothesis is in disagreement with phylogenetic signals produced by other data types. To resolve this conflict in signals between mitochondrial and nuclear (as well as morphological) data, we proposed that Cymothoidae + Corallanidae probably underwent an additional ROI event, which caused another inversion of skews, and thus resulted in skew values homoplastic with the Asellota and other crustaceans [1]. Following previous evidence that asymmetrical skews can impede phylogenetic reconstruction [14,18], we proposed that these multiple inversions of mitochondrial skews in isopods cause conflicting signals between the mitogenomic and nuclear/morphological data, largely by producing artefactual clustering of species exhibiting homoplastic skews at the base of the isopod clade.

As inverted skews are reflected on the composition of all mitochondrial genes, LBA artefacts are observable regardless of the number of mitochondrial genes used for the phylogenetic analysis [1,3,9,11,19,20]. Furthermore, architecture-driven mutational pressures can be reflected on the amino acid composition as well [21,22], which explains why the use of amino acid sequences also failed to resolve the LBA [1]. Although these observations led us to conclude that mitogenomic molecular data are not a suitable tool to resolve the phylogeny of Isopoda, our conclusions may have been biased by two methodological problems: poor taxon sampling and the inclusion of outgroups in our phylogenetic analyses. Furthermore, the existence of such a clearly identifiable LBA artefact presents a good model to test the impacts of different methodological approaches on the magnitude of LBA [23]. As regards the taxon sampling, in the previous study, we had only three species exhibiting D-I skews at disposal, and we removed several incomplete isopod mitogenomes from the dataset to maximize the amount of characters included in the analyses. As there is evidence that improved taxonomic sampling is much more efficient at resolving LBA artefacts than adding characters [23,24], for this study, we sequenced the complete mitogenome of an additional isopod species expected to exhibit a D-I skew, and used all available isopod mitogenomes for the phylogenetic analyses. In addition, outgroup taxa usually exhibit long branches, which may cause LBA artefacts with long-branched ingroup taxa [23,25]. As other Malacostraca and D-I isopod taxa (including Asellota) exhibit homoplastic skews, the inclusion of these outgroup taxa may also have skewed the results of our analyses and resulted in the D-I taxa clustering at

the base of the isopod clade. To test the impact of outgroups on the observed LBA artefact, we conducted phylogenetic analyses using three different outgroup selection strategies (including a dataset without an outgroup). Furthermore, isopods exhibit an exceptionally destabilized mitogenomic architecture, with all sequenced species exhibiting unique gene orders [5,20], so they might present a suitable model system to study discontinuity in the dynamics of evolution of mitochondrial genomic architecture [11,26–28]. Therefore, for this study, we sequenced the mitogenome of another isopod species belonging to the clade of parasitic cymothoids that exhibit D-I skews, *Asotana magnifica* Thatcher, 1988 (Cymothoidae). We conducted comparative mitogenomic architecture analyses, with focus on skew patterns, and conducted phylogenetic analyses using several different datasets. This allowed us to test the following working hypotheses: (1) D-I skews previously observed in three isopod species were a result of an additional ROI event that occurred in the common ancestor of Cymothoidae and Corallanidae families, and all species belonging to these families, including *A. magnifica*, should exhibit D-I skews; (2) in agreement with the high rate of mitogenomic architecture evolution in isopods, the mitogenome of *A. magnifica* should exhibit a unique mitogenomic architecture (gene order); (3) due to strong asymmetrical skews of isopod mitogenomes, improved taxonomic sampling should fail to resolve skew-driven phylogenetic artefacts (LBA), and (4) due to homoplastic skews between Asellota and Cymothoidae + Corallanidae, LBA artefact will remain pronounced regardless of the outgroup selection strategy.

# 2. Methods

## 2.1. Sample and identification

A single adult female *A. magnifica* specimen was collected on 18 January 2017 from the buccal cavity of an unidentified live piranha (*Serrasalmus* sp.) specimen (origin unknown) from a private freshwater aquarium in Wuhan, China. The parasite was kept alive in water for two days to ensure that it was starved and then stored in 75% ethanol at 4°C. It was photographed (figure 1), and morphologically identified under a dissecting microscope according to the original description of this species [29]. The host fish was anaesthetized with MS-222 during the removal of the parasite, and otherwise handled in accordance with the guidelines for the care and use of animals for scientific purposes set by the Ministry of Science and Technology, Beijing, China (no. 398, 2006). As the remaining procedures involved an unregulated parasitic invertebrate, no special permits were required to retrieve and process the sample.

## 2.2. Genome sequencing and assembly

Genome sequencing and assembly were conducted as described before [11,20,28]. Briefly, after washing the sample in distilled water, DNA was isolated from the complete specimen using AidLab DNA extraction kit (AidLab Biotechnologies, Beijing, China). Nine primer pairs (table 1) used to amplify and sequence the entire mitogenome were designed to match conserved regions of mitochondrial genes (assessed using available orthologues) and to overlap by approximately 100 bp. PCR reaction mixture (50 µl): 5 U µl$^{-1}$ TaKaRa LA Taq polymerase (TaKaRa, Japan), 10 × LATaq Buffer II, 2.5 µM dNTP mixture, 0.2–1.0 µM each primer and 60 ng DNA template. Below-mentioned conditions were followed: denaturation 98°C/2 min, 40 cycles of 98°C–10 s, 50°C–15 s and 68°C–1 min kb$^{-1}$. When the product was not specific enough, PCR conditions were optimized by increasing the annealing temperature and decreasing the number of cycles. PCR products were sequenced using Sanger method and the same set of primers. After quality proofing via the visual inspection of electropherograms and identity confirmation using BLAST [30], the mitogenome was assembled manually using DNASTAR v. 7.1 [31]. We made sure that overlaps were identical, the mitogenome circular, and that no *numt*s [32] were incorporated. ORFs for protein-coding genes (PCGs) were approximately located using DNASTAR and then fine-tuned according to the orthologous sequences using BLAST and BLASTx. The two ribosomal RNAs were also annotated via a comparison with orthologues. tRNAs were annotated using tRNAscan [33] and ARWEN [34] tools. PhyloSuite [35] was used to parse and extract the annotation recorded in a Word (Microsoft Office) document and to generate the file for submission to GenBank.

## 2.3. Comparative mitogenomic and phylogenetic analyses

All 28 available isopod mitogenomes were retrieved from the GenBank for comparative and phylogenetic analyses. We conducted analyses on three different datasets, all of which comprised all 29

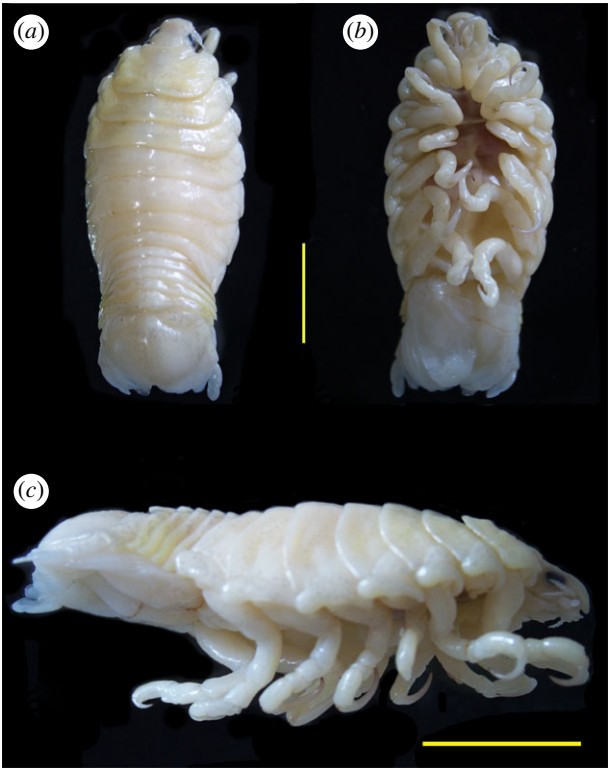

**Figure 1.** Photographs of the *Asotana magnifica* (female) specimen used for DNA extraction. (*a*) dorsal view; (*b*) ventral view; (*c*) lateral view. Scale (yellow bar) = 5 mm.

**Table 1.** Primers used for amplification and sequencing of the mitochondrial genome of *Asotana magnifica*.

| fragment | gene or region | primer name | sequence (5′-3′) | length (bp) |
|---|---|---|---|---|
| F1 | ATP6 | SSF1 | GTCAACNTTWAGTAGTCYTC | 787 |
| | | SSR1 | GCTACTDCTGTTTCTAGG | |
| F2 | ATP6-12S | SSF2 | TCTCATTCAGCCTCCTAAAC | 2977 |
| | | SSR2 | CTTTTACTACCTTGTCTTG | |
| F3 | 12S | SSF3 | CTGACANAACADGTGCCAGC | 551 |
| | | SSR3 | CTATGTTACGACWTGCCTCY | |
| F4 | 12S-CYTB | SSF4 | CAAAGATAAACTTTTACCTCAGG | 1439 |
| | | SSR4 | GTTATGAAGGATGCTGTTGGG | |
| F5 | CYTB | SSF5 | GAATATGAACAGGTGTTCTC | 420 |
| | | SSR5 | TGGMGTATGTTCTNCCTTGG | |
| F6 | CYTB-NAD4 L | SSF6 | GAGTTACTCCTCTTACAGC | 3881 |
| | | SSR6 | GTCTCATTGTTAGCTTTGGAG | |
| F7 | NAD4 L-16S | SSF7 | CCATATTCATTTTATCTCTTCC | 1113 |
| | | SSR7 | CGTAATCGTATTTGGGAGTTC | |
| F8 | 16S | SSF8 | GATAGAAAHCAACCTNGCTTAC | 490 |
| | | SSR8 | GTAGVCTCTGTTCAATGATGAC | |
| F9 | 16S-ATP6 | SSF9 | GGGTCTTGTCGTCCCTTTAG | 4553 |
| | | SSR9 | TCAATAGGTGTAAGGTAG | |

(28 + *A. magnifica*) isopod mitogenomes, but differed in the selection of outgroups. The largest dataset (39 mitogenomes) comprised the same outgroups as in our previous study: nine mitogenomes belonging to five other Malacostraca orders (to allow us to determine the sister group to Isopoda),

and a basal arthropod *Limulus polyphemus* [22]. To assess whether these 10 outgroup species might be producing a strong 'gravitational' effect on other species exhibiting homoplastic skews (D-I skew species and *A. aquaticus*), thus exacerbating the LBA, we tested the performance of a dataset with only one outgroup species (30 mitogenomes). Since Amphipoda are considered to be the most prominent contender for the position of the sister group to Isopoda [1,3], *Eulimnogammarus cyaneus* [36] was chosen for this task. Finally, to detect whether this outgroup merely roots the isopod dataset, or if it simultaneously alters its topology [23], we also conducted analyses on a dataset without an outgroup. As we discovered previously that the CAT-GTR model implemented in PhyloBayes in combination with concatenated amino acid sequences of all 13 PCGs was the most successful method in alleviating the LBA [1], we used this methodological approach to assess the performance of these datasets. PhyloSuite (and its plug-in programs) was used to retrieve data from the GenBank, semi-automatically re-annotate ambiguously annotated tRNA genes with the help of the ARWEN output, extract the mitogenomic features, translate genes into amino acid sequences, generate comparative tables, calculate skews, align the 13 genes in batch mode using the accurate L-INS-i strategy and normal alignment mode in MAFFT [37], concatenate alignments, infer the best data partitioning scheme and select best-fit evolutionary models for each partition using Akaike Information Criterion implemented in PartitionFinder2 [38], and prepare input files for other programs. Phylogenetic analyses were conducted using concatenated amino acid sequences of all 13 PCGs and two different algorithms: Bayesian inference analyses were conducted using MrBayes 3.2.6 [39] with default settings and $5 \times 10^6$ generations; CAT-GTR model analyses were conducted using PhyloBayes-MPI 1.7a [40], available from the CIPRES server [41]. Analyses were run with default parameters (burnin = 500, invariable sites automatically removed from the alignment, two MCMC chains) and automatically stopped when the conditions considered to indicate a good run were reached: maxdiff < 0.1 and minimum effective size > 300 (PhyloBayes manual). Phylograms and gene orders were visualized in iTOL [42], and annotated using files generated by PhyloSuite.

# 3. Results and discussion

## 3.1. Architecture and characteristics of the mitogenome of Asotana magnifica

The circular mitochondrial genome of *A. magnifica* is 14 435 bp long, with the A + T content of 68.1%. It possesses the standard 13 PCGs and two rRNA genes (*12S* and *16S*), but only 19 tRNA genes, as three tRNA genes could not be identified with certainty: *trnC*, *trnI* and *trnE* (table 2). The mitogenome exhibits a unique gene order (figure 2). A 475 bp long putative CR was found in the ancestral arthropod position [22], adjacent to the *12S* gene. Twelve genes were encoded on the minus strand. Most PCGs exhibit sizes and start/stop codons standard for isopods (electronic supplementary material, File: worksheet B). As standard for mitogenomes of most animals [43], including isopods [11,16], the mitogenome of *A. magnifica* is also highly compact: we identified 12 non-coding regions (NCRs; also referred to as intergenic spacers), only four of which were larger than 10 bp, and 14 gene overlaps. Although exceptionally large overlaps were identified in mitogenomes of some isopods from the genus *Armadillidium* (Oniscidea) [44], there is no indication of such overlaps in *A. magnifica*. The largest (both 7 bp) were found between *atp6/8* and *nad4/4 L* genes, but the overlap between these genes is conserved in many metazoan lineages [45], including the isopods [11].

Although incomplete tRNA sets are common in Isopoda [5,10,11,20], to make sure that we did not oversee the three non-identified tRNA genes, we additionally checked both strands of all four large NCRs, including ≈10–20 bp overlaps with neighbouring genes using ARWEN and tRNAscan. We could not identify complete tRNAs (i.e. sequences that can be folded into a functional cloverleaf structure), but we found highly conserved central segments of *trnC* and *trnE* genes (including anti-codons) in the two NCRs that correspond to the locations where the two genes were identified in related species (figure 2): *nad2*-34bpNCR-*trnY* and *trnL1*-41bpNCR-*trnL2*, respectively. The position of *trnI* is too variable to identify it with confidence. As extensive post-transcriptional editing of tRNA genes has been reported in some *Armadillidium* species [44], we speculate that these fragments of tRNA genes might actually be transcribed and subsequently edited into functional tRNA genes. This observation may explain why apparently incomplete tRNA sets are common in isopods and deserves further research. These missing tRNA genes also explain some (but not all) of the apparent gene order rearrangements.

The high compactness of isopod mitogenomes is very intriguing in the light of the rapid gene order evolution in this lineage. Often, taxa that exhibit high rates of mitochondrial architecture evolution also

**Table 2.** Organization of the mitochondrial genome of *Asotana magnifica*. IGR is intergenic region, where a negative value indicates an overlap. CR is (putative) control region.

| gene | position | | | | codon | | | |
| | from | to | size | IGR | start | stop | anti-codon | strand |
|---|---|---|---|---|---|---|---|---|
| trnL1 | 1 | 58 | 58 | | | | TAG | − |
| trnL2 | 100 | 160 | 61 | 41 | | | TAA | − |
| trnS1 | 152 | 216 | 65 | −9 | | | TCT | − |
| trnW | 218 | 269 | 52 | 1 | | | TCA | − |
| cytb | 274 | 1399 | 1126 | 4 | ATT | T | | − |
| trnT | 1400 | 1456 | 57 | | | | TGT | − |
| nad5 | 1456 | 3147 | 1692 | −1 | ATA | TAA | | + |
| trnF | 3140 | 3197 | 58 | −8 | | | GAA | + |
| trnH | 3190 | 3247 | 58 | −8 | | | GTG | + |
| nad4 | 3247 | 4557 | 1311 | −1 | CTA | CAT | | + |
| nad4 L | 4551 | 4850 | 300 | −7 | TTA | AAT | | + |
| trnP | 4851 | 4908 | 58 | | | | TGG | − |
| nad6 | 4910 | 5395 | 486 | 1 | ATA | TAA | | + |
| trnS2 | 5394 | 5454 | 61 | −2 | | | TGA | + |
| rrnL | 5455 | 6533 | 1079 | | | | | − |
| trnQ | 6534 | 6588 | 55 | | | | TTG | − |
| trnM | 6580 | 6644 | 65 | −9 | | | CAT | + |
| nad2 | 6645 | 7628 | 984 | | ATT | TAA | | + |
| trnY | 7663 | 7723 | 61 | 34 | | | GTA | − |
| cox1 | 7724 | 9259 | 1536 | | ATG | TAA | | + |
| cox2 | 9329 | 10 006 | 678 | 69 | ATC | TAA | | + |
| trnK | 10 005 | 10 065 | 61 | −2 | | | TTT | + |
| trnD | 10 073 | 10 115 | 43 | 7 | | | GTC | + |
| atp8 | 10 125 | 10 277 | 153 | 9 | ATA | TAA | | + |
| atp6 | 10 271 | 10 937 | 667 | −7 | ATG | T | | + |
| cox3 | 10 938 | 11 723 | 786 | | ATG | TAA | | + |
| trnR | 11 722 | 11 777 | 56 | −2 | | | TCG | + |
| trnG | 11 781 | 11 829 | 49 | 3 | | | TCC | + |
| nad3 | 11 829 | 12 179 | 351 | −1 | ATA | TAA | | + |
| trnA | 12 183 | 12 220 | 38 | 3 | | | TGC | + |
| trnV | 12 218 | 12 274 | 57 | −3 | | | TAC | + |
| nad1 | 12 274 | 13 194 | 948 | −1 | ATT | TAA | | − |
| trnN | 13 196 | 13 256 | 61 | 1 | | | GTT | + |
| rrnS | 13 257 | 13 959 | 703 | | | | | + |
| CR | 13 960 | 14 435 | 475 | | | | | |

exhibit multiple expanded NCRs, sometimes spanning hundreds and even thousands of bases [28,46,47]. In some cases, these are believed to be a consequence of the tandem-duplication-random-loss (TDRL) rearrangement mechanism, wherein pseudogenes and non-coding DNA are generated and then again lost over the evolutionary time [43,47–50]. Therefore, high number of gene order rearrangements between closely related taxa and high compactness of their mitogenomes strongly suggest that TDRL

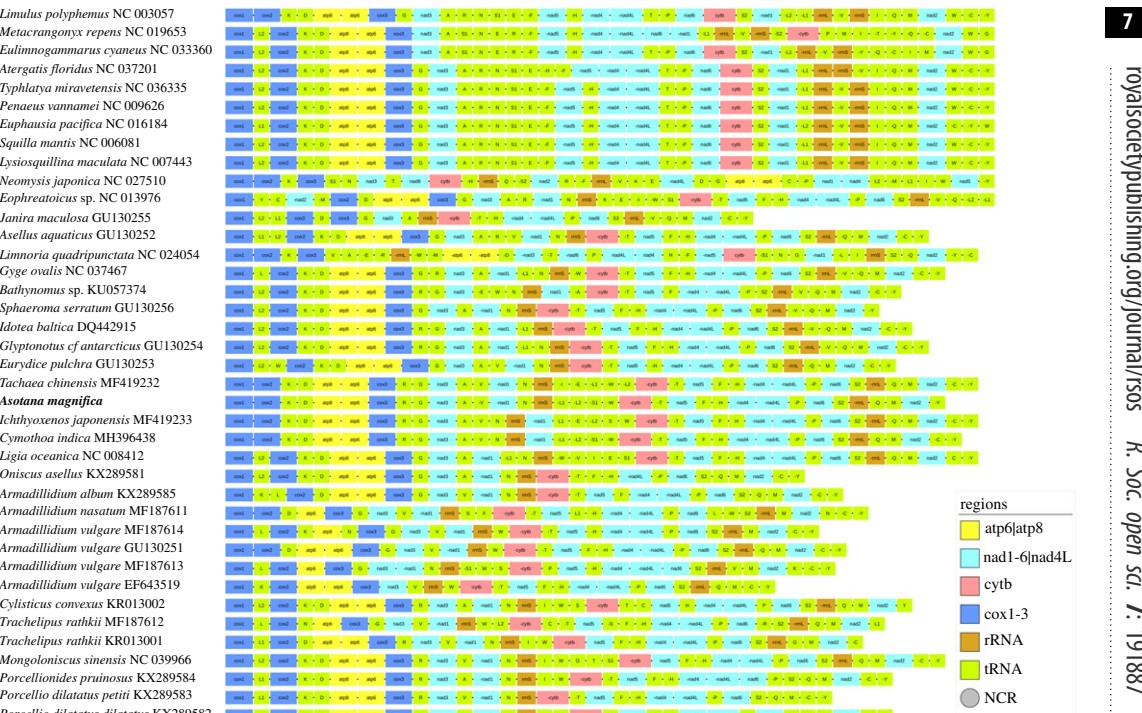

**Figure 2.** Gene orders of Isopoda (and selected crustaceans). GenBank numbers of sequences are shown next to species' names, taxonomic details are provided in the electronic supplementary material, File and the newly sequenced *Asotana magnifica* is bolded. Colour legend is shown in the figure.

mechanism is not the most parsimonious explanation for the high rate of gene order rearrangements in isopods. Mitogenome fragmentation into multipartite genomes has been observed in many other animal taxa [51], including crustaceans [52], and some isopod species belonging to the suborders Oniscidea and Asellota possessing unusual and highly destabilized mitogenomic organization, including linearization and dimerization [53,54]. Therefore, it is possible that destabilized architecture of isopod mitogenomes may have facilitated sporadic linearization and fragmentation events [11]. As linearization events are believed to facilitate homologous (intramitochondrial) DNA recombination, and that most gene (and replication origin) inversions found in vertebrates can be attributed to this mechanism [49,55], we hypothesize that this mechanism is also the best explanation for the highly elevated frequency of mitochondrial architecture rearrangements in the evolutionary history of isopods.

## 3.2. Skew patterns in the mitogenome of *Asotana magnifica*

In general, there is an increasing evidence that the evolution of mitogenomes, especially base composition skews, is to a large extent driven by nonadaptive architecture-associated pressures, predominantly associated with mitochondrial replication [13,56]. In concord with other three available Cymothoidae + Corallanidae species [1], *A. magnifica* also exhibits inverted skews on the majority strand in relation to most other isopods: positive AT skew (0.111) and negative GC skew (−0.423) (figure 3; electronic supplementary material, File: worksheet D). The magnitude of skews is higher on the minority (−) than on the majority (+) strand. As mitogenomic GC skews are generally greater on the leading strand than on the lagging strand [57], this observation also indirectly supports the hypothesis of an additional ROI in this mitogenome. All genes encoded on the majority strand exhibited negative GC skews, and most of them exhibited positive AT skews (exceptions were only *atp8*, *cox1*, *cox3*, *nad2* and the second codon position of concatenated PCGs). GC skews by codon positions of genes on the majority strand exhibited a steep increase from first to third codon (−0.122 to −0.769), but a decrease in the GC skewness of second codon was exhibited by the minority strand genes (first = 0.547, second = 0.323, third = 0.825). As mitochondrial genes generally evolve under a strong purifying selection [58], nonadaptive hydrolytic deamination of bases has far more pronounced impacts on the third codon position (synonymous mutations) of PCGs than on the first two codon positions [59]. The decreased magnitude of the

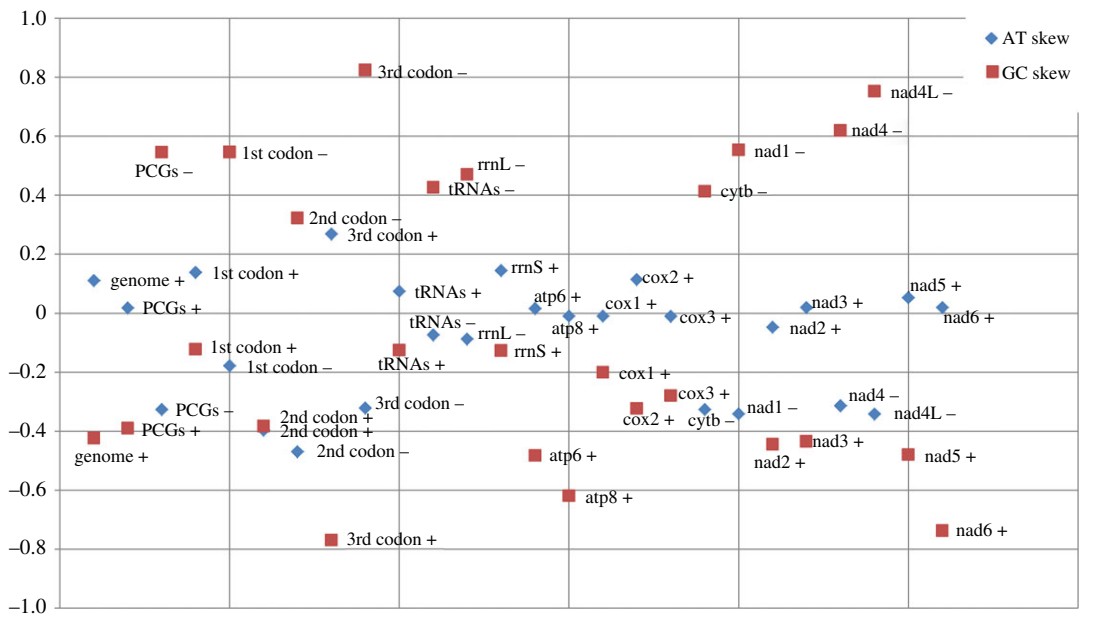

**Figure 3.** Skews in the mitogenome of *Asotana magnifica*. +/− indicate majority/minority strand, respectively. 'codon' and 'PCGs' values were calculated for concatenated PCGs (protein-coding genes) encoded on the same strand.

second codon skew, as well as lesser magnitude of AT skews compared to GC skews, can be explained by the strong selection for codons containing T at the second codon position in membrane-spanning protein segments, which must be hydrophobic to ensure the conformational stability [21,59,60]. Along with the second codon, mitochondrial genes known to generally evolve under the strongest purifying selection, such as the *cox* family, exhibited the lowest absolute GC skews (i.e. closest to 0), whereas genes known to evolve at comparatively relaxed selection constraints, e.g. *atp8*, *nad4 L* and *nad6* [61–63], exhibited the highest absolute skews. Therefore, although base composition skewness of the mitogenome of *A. magnifica* is primarily driven by nonadaptive architectural factors, adaptive purifying selection has pronounced effects on certain elements of the mitogenome.

## 3.3. Phylogenetic analyses: impacts of taxon sampling and outgroup selection on LBA

Previously, we proposed that species from Cymothoidae and Corallanidae families underwent an additional ROI event, which resulted in a D-I skew, and we showed that homoplastic skews between Asellota and Cymothoidae + Corallanidae produce a phylogenetic artefact of these species clustering together at the base of the entire isopod clade [1]. This topology is in disagreement with other data types (nuclear and morphological) which suggest that Cymothoidae + Corallanidae are one of the most derived (newest) isopod clades [2–11] (also see 18S dataset phylogram in the electronic supplementary material, File: worksheet C). Furthermore, we argued that compositional heterogeneity causes additional instability in the position of several 'rogue' taxa, such as *Ligia oceanica* (Ligiidae), *Eurydice pulchra* (Cirolanidae) and *Limnoria quadripunctata* (Limnoriidae). However, for those analyses, we only had three species exhibiting D-I skews at disposal, we did not use six of the available mitogenomes and we used a large number of outgroups to stabilize the topology. As both taxon sampling and outgroup choice can produce great effects on the topology [24,25,64,65], for this study, we sequenced an additional species liable to exhibit a D-I skew, used all 28 isopod mitogenomes available in the GenBank (completeness shown in the electronic supplementary material, File: worksheet A) and tested impacts of different outgroup sampling strategies.

First, we conducted a standard Bayesian analysis on partitioned datasets to corroborate that homogeneous models are not sensitive to additional data and outgroup selection. The artefact of homoplastic skew taxa belonging to Asellota (*A. aquaticus* and *J. maculosa*) and Cymothoida (four D-I skew species) forming a monophyletic clade was reproduced by all three analyses (electronic supplementary material, File: worksheet C). The topology inferred using the same heterogeneous model (CAT-GTR) and outgrouping strategy as in our previous study (10 outgroups; figure 4*a*) was highly congruent with the one inferred in the previous study [1], apart from a minor rearrangement

**Figure 4.** Impact of outgrouping strategy on the phylogenetic reconstruction in Isopoda. (*a*) A dataset comprising ten outgroup taxa (topmost taxa, ending with *Neomysis japonica*). (*b*) A dataset with one outgroup (*Eulimnogammarus cyaneus*; branch cropped). (*c*) A dataset without an outgroup. All phylograms were reconstructed using CAT-GTR model and concatenated amino acid sequences of 13 mitochondrial protein-coding genes. Posterior support values are shown next to nodes. The scale bar corresponds to the estimated number of substitutions per site. GenBank numbers and taxonomic details are provided in figure 2 and electronic supplementary material, File. Clades containing isopod taxa with negative GC skews are marked with a bolded '-' character at the base.

observed in the position of *Bathynomus* sp. Therefore, we can conclude that improved sampling (increased number of taxa) did not affect the topology. However, the CAT-GTR analyses were sensitive to the outgroup selection strategy. A fairly congruent topology was inferred using a dataset without an outgroup (here rooted using *Eophreatoicus* sp. to make the topologies more comparable; figure 4*b*): the two topologies differed in the arrangement of Asellota and two rogue [1] taxa, *L. quadripunctata* and *Gyge ovalis*. However, the dataset with a single outgroup produced a similar LBA artefact as homogeneous models: the D-I skew species at the base of the isopod clade (figure 4*c*). Therefore, we can conclude that the use of a single outgroup exacerbated the LBA, whereas the use of a large number of outgroups and complete exclusion of outgroups ameliorated it. Intriguingly, some mitochondrial phylogenomics studies of datasets expected to exhibit compositional heterogeneity found that the outgroup choice does not affect the topology at all [66]. We hypothesize that these effects might be comparatively strongly pronounced in isopods due to exceptionally high compositional heterogeneity of their mitogenomes.

As the two 'better' topologies also exhibited several polytomies, deeply paraphyletic Cymothoida, and instability in the position of several rogue taxa (*L. quadripunctata*, *G. ovalis*, and *Bathynomus* sp.), we can conclude that improved taxon sampling and outgroup selection did not manage to fully stabilize the topology. Despite this, a few observations deserve to be highlighted. As regards the rogue taxa, whose positions often vary among studies [1–3,5,9,10,67], our analyses offer additional evidence that Ligiidae (here represented by *L. oceanica*) are the most primitive family in the Oniscidea suborder [1,68]. The position of another rogue species, *E. pulchra* (Cymothoida: Cirolanidae), was relatively stable, at the base of the D-I skew clade (Corallanidae + Cymothoidae). Despite this, the suborder Cymothoida remained paraphyletic due to unorthodox positions of *Bathynomus* sp. (Cirolanidae) and *G. ovalis* (Bopyridae). It would be interesting to sequence further samples of these two (and closely related) species and attempt to assess whether their rogue behaviour is caused by compositional heterogeneity or by taxonomically misidentified samples. Finally, the position of the rogue Limnoriidae clade (*L. quadripunctata*) was unresolved (polyphyly), but generally at the base of the central catch-all [1] isopod clade.

# 4. Conclusion

Our analyses corroborated three of our working hypotheses. (1) The mitogenome of *A. magnifica* also exhibited a D-I skew, thereby offering additional evidence for the proposed ROI event in the common ancestor of Cymothoidae and Corallanidae families. Although base composition skewness of the mitogenome of *A. magnifica* is primarily driven by nonadaptive architectural factors, adaptive purifying selection has pronounced effects on certain elements of the mitogenome. (2) The mitogenome exhibited a unique mitogenomic architecture. This corroborates that isopods are undergoing exceptionally accelerated gene order evolution, and that they may be a good model to study the evolution of mitogenomic architecture. We hypothesize that linearization events followed by homologous (intramitochondrial) DNA recombination is most likely to be the mechanism underlying this high frequency of mitochondrial architecture rearrangements (including the ROI events) in the evolutionary history of Isopoda. (3) Due to asymmetrical skews of isopod mitogenomes, improved taxonomic sampling failed to fully resolve the skew-driven phylogenetic artefacts. Finally, as regards the fourth hypothesis, LBA artefacts and topological instability remained pronounced regardless of the outgroup selection strategy, so we can tentatively accept it. However, our results offer some important methodological implications, namely, the inclusion of a single outgroup strikingly exacerbated the LBA artefact, whereas both the inclusion of a large number of outgroups (10) belonging to different Malacostraca orders and complete exclusion of outgroups notably ameliorated the LBA.

Limitations of our study are largely in the incomplete sampling with regard to the currently accepted isopod taxonomy, especially the absence of two isopod suborders from our dataset (Phoratopidea and Microcerberidea), rogue Limnoriidea suborder represented by only one species, and the already discussed poor sampling of some other rogue cymothoid taxa. However, as five out of seven accepted isopod suborders [69] were included in our analyses, and as the two missing suborders are relatively small (one and two families, respectively), we argue that their inclusion is highly unlikely to fully resolve these LBA artefacts. Despite of the fact that mitochondrial molecular data are not a suitable tool for resolving the phylogeny of this order, highly destabilized GOs, and multiple skew inversions (ROI) make isopod mitogenomes exceptionally interesting from the perspective of mitochondrial architecture evolution. In addition, herein we identified highly conserved central segments of *trnC*

and *trnE* genes (including anti-codons) in the two NCRs that correspond to the locations where the two genes were identified in related species and proposed that these fragments of tRNA genes might actually be transcribed and subsequently edited into functional tRNA genes. Therefore, isopods may also present a good model to study this intriguing [70] evolutionary phenomenon. Improved sampling of taxa is also necessary to better understand the skew patterns in isopods and indirectly infer the evolutionary history of ROI events in this crustacean order. Particularly interesting from this aspect is the non-represented suborder Microcerberidea, which is believed to be the sister group of Asellota [71], so we hypothesize that it should exhibit the common crustacean skew pattern. Also, mitogenomes are available only for three families of the superfamily Cymothooidea, two of which exhibit D-I skews. Therefore, it would be necessary to sequence representatives of all six remaining families in order to ascertain whether the D-I skew taxa are indeed monophyletic and infer the exact evolutionary history timing of the additional ROI event.

Ethics. The host fish was anaesthetized with MS-222 during the removal of the parasite and otherwise handled in accordance with the guidelines for the care and use of animals for scientific purposes set by the Ministry of Science and Technology, Beijing, China (no. 398, 2006). As the remaining procedures involved an unregulated parasitic invertebrate, no special permits were required to retrieve and process the sample.

Data accessibility. The sequenced mitogenome is deposited in GenBank (RefSeq) under the accession number NC_042824. Other datasets supporting this article have been uploaded as electronic supplementary material.

Authors' contributions. H.Z., I.J., D.Z. and G.-T.W. conceived and designed the study; H.Z., I.J., D.Z., R.C., C.-J.H., W.-X.L. and M.L. acquired, analysed and interpreted the data; H.Z. and I.J. drafted the article; D.Z., R.C., C.-J.H., W.-X.L., M.L. and G.-T.W. revised the article critically for important intellectual content; all authors gave final approval for publication and agree to be accountable for all aspects of the work.

Competing interests. We declare we have no competing interests.

Funding. This work was supported by the National Natural Science Foundation of China (grant no. 31970408) and Earmarked Fund for China Agriculture Research System (grant no. CARS-45-15).

Acknowledgements. We would like thank Kerry A. Hadfield, Roberto Ardovini and Panakkool Thamban Aneesh for helping us to identify the *A. magnifica* specimen. We also thank two anonymous referees for investing time and expertise into reviewing our manuscript.

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
