## [Reviewer comments · Royal Society Open Science]

Review History

RSOS-191887.R0 (Original submission)

Review form: Reviewer 1

Is the manuscript scientifically sound in its present form?

Yes

Are the interpretations and conclusions justified by the results?

Yes

Is the language acceptable?

Yes

Do you have any ethical concerns with this paper?

No

Have you any concerns about statistical analyses in this paper?

No

Recommendation?

Accept with minor revision (please list in comments)

Comments to the Author(s)

Zou et al. sequenced the complete mitogenome of *Asotana magnifica*, and conducted comparative mitogenomic architecture analyses, with focus on skew patterns, and conducted phylogenetic analyses using several different datasets. The work on mitochondrial skews description is thorough, and the paper is well written.

Some minor points the author should address in the revised version are listed as below.

Page 4 Line10-11: you say “different datasets.....often producing starkly contradictory phylogenetic hypotheses”. It's better to have the reference behind it.

Page 4 Line 18: “This is to”, how does “this” refer to? In this section, you use many “this” word. If used incorrectly, it can confuse the reader.

Page 4 Line 50: What is “ORI” ?

Page 10 Line 32: Here you use “only 12 non-coding regions”, but before and after the article did not explain clearly, how many non-coding areas are considered normal?

Page 13 Line 42-48: “As mitochondrial genes.....the third codon position (synonymous mutations)”. It's better to have the reference behind it.

Review form: Reviewer 2 (Jianmei An)

Is the manuscript scientifically sound in its present form?

Yes

Are the interpretations and conclusions justified by the results?

Yes

Is the language acceptable?

Yes

Do you have any ethical concerns with this paper?

No

Have you any concerns about statistical analyses in this paper?

No

Recommendation?

Accept with minor revision (please list in comments)

Comments to the Author(s)

The manuscript is perfect and suitable for publication in your journal. But there are minor revision need to be done.

1. Page 4, Line 3-39; The known exception is not only one species, but two species, They are *Asellus aquaticus* (GC-Skew=-0.122) and *Janira maculosa* (GC-Skew=-0.026);
2. Page 17, Line 59; Reference 63 is 2018; should use the newest literature (2019).

Decision letter (RSOS-191887.R0)

02-Jan-2020

Dear Dr Jakovlic

On behalf of the Editors, I am pleased to inform you that your Manuscript RSOS-191887 entitled "Architectural instability, inverted skews, and mitochondrial phylogenomics of Isopoda: outgroup choice affects the long-branch attraction artefacts" has been accepted for publication in Royal Society Open Science subject to minor revision in accordance with the referee suggestions. Please find the referees' comments at the end of this email.

The reviewers and handling editors have recommended publication, but also suggest some minor revisions to your manuscript. Therefore, I invite you to respond to the comments and revise your manuscript.

- Ethics statement

- Data accessibility

<http://datadryad.org/submit?journalID=RSOS&manu=RSOS-191887>

- Competing interests

- Authors' contributions

AB carried out the molecular lab work, participated in data analysis, carried out sequence alignments, participated in the design of the study and drafted the manuscript; CD carried out

the statistical analyses; EF collected field data; GH conceived of the study, designed the study, coordinated the study and helped draft the manuscript. All authors gave final approval for publication.

- Acknowledgements

- Funding statement

Because the schedule for publication is very tight, it is a condition of publication that you submit the revised version of your manuscript before 11-Jan-2020. Please note that the revision deadline will expire at 00.00am on this date. If you do not think you will be able to meet this date please let me know immediately.

If your manuscript is newly submitted and subsequently accepted for publication, you will be asked to pay the article processing charge, unless you request a waiver and this is approved by Royal Society Publishing. You can find out more about the charges at <https://royalsocietypublishing.org/rsos/charges>. Should you have any queries, please contact openscience@royalsociety.org.

on behalf of Dr David Ferrier (Associate Editor) and Kevin Padian (Subject Editor)
openscience@royalsociety.org

Associate Editor Comments to Author (Dr David Ferrier):

Both reviewers have made a handful of suggestions for some minor revisions, which should be accommodated in order to make this manuscript acceptable for publication.

Reviewer comments to Author:
Reviewer: 1

Comments to the Author(s)

Zou et al. sequenced the complete mitogenome of *Asotana magnifica*, and conducted comparative mitogenomic architecture analyses, with focus on skew patterns, and conducted phylogenetic analyses using several different datasets. The work on mitochondrial skews description is thorough, and the paper is well written.

Some minor points the author should address in the revised version are listed as below.

Page 4 Line10-11: you say “different datasets.....often producing starkly contradictory phylogenetic hypotheses”. It's better to have the reference behind it.

Page 4 Line 18: “This is to”, how does “this” refer to? In this section, you use many “this” word. If used incorrectly, it can confuse the reader.

Page 4 Line 50: What is “ORI” ?

Page 10 Line 32: Here you use "only 12 non-coding regions", but before and after the article did not explain clearly, how many non-coding areas are considered normal?

Page 13 Line 42-48: "As mitochondrial genes.....the third codon position (synonymous mutations)". It's better to have the reference behind it.

Reviewer: 2

Comments to the Author(s)

The manuscript is perfect and suitable for publication in your journal. But there are minor revision need to be done.

1. Page 4, Line 3-39; The known exception is not only one species, but two species, They are *Asellus aquaticus* (GC-Skew=-0.122) and *Janira maculosa* (GC-Skew=-0.026);
2. Page 17, Line 59; Reference 63 is 2018; should use the newest literature (2019).

Author's Response to Decision Letter for (RSOS-191887.R0)

See Appendix A.

Decision letter (RSOS-191887.R1)

14-Jan-2020

Dear Dr Jakovlic,

It is a pleasure to accept your manuscript entitled "Architectural instability, inverted skews, and mitochondrial phylogenomics of Isopoda: outgroup choice affects the long-branch attraction artefacts" in its current form for publication in Royal Society Open Science.

Due to rapid publication and an extremely tight schedule, if comments are not received, your paper may experience a delay in publication. Royal Society Open Science operates under a continuous publication model. Your article will be published straight into the next open issue and this will be the final version of the paper. As such, it can be cited immediately by other researchers. As the issue version of your paper will be the only version to be published I would

advise you to check your proofs thoroughly as changes cannot be made once the paper is published.

on behalf of Dr David Ferrier (Associate Editor) and Kevin Padian (Subject Editor)
openscience@royalsociety.org

Appendix A

Associate Editor Comments to Author (Dr David Ferrier):

Both reviewers have made a handful of suggestions for some minor revisions, which should be accommodated in order to make this manuscript acceptable for publication.

R: We appreciate the time and expertise that you and reviewers invested into our manuscript. The comments and critiques were constructive and helpful. We addressed all problems as best as the available literature allowed us. We also proofread the entire manuscript once again and made a few changes in places where we felt that there is room for improved clarity. All changes can be reviewed in the 'track-change' version of the manuscript.

Reviewer comments to Author:

Reviewer: 1

Comments to the Author(s)

Zou et al. sequenced the complete mitogenome of *Asotana magnifica*, and conducted comparative mitogenomic architecture analyses, with focus on skew patterns, and conducted phylogenetic analyses using several different datasets. The work on mitochondrial skews description is thorough, and the paper is well written. Some minor points the author should address in the revised version are listed as below.

Page 4 Line10-11: you say "different datasets.....often producing starkly contradictory phylogenetic hypotheses". It's better to have the reference behind it.

R: We discuss specific problems in the text immediately following the statement, and support it with a large number of references. Following your objection, we added three references that present a partial, but relatively good introduction into the multitude of phylogenetic hypotheses and topologies produced by different studies: "...often producing starkly contradictory phylogenetic hypotheses [1–3]."

Page 4 Line 18: "This is to", how does "this" refer to? In this section, you use many "this" word. If used incorrectly, it can confuse the reader.

R: Following your comment, we edited the entire section for clarity.

Page 4 Line 50: What is "ORI" ?

R: We apologise for this mistake. We initially used ORI (origin of replication inversion) abbreviation, but then decided to change it to ROI (replication origin inversion) later. We fixed it.

Page 10 Line 32: Here you use "only 12 non-coding regions", but before and after the article did not explain clearly, how many non-coding areas are considered normal?

R: Unlike the nuclear genome, which is mostly comprised of non-coding DNA, mitogenomes are usually highly compact, with very little non-coding DNA [4,5]. With 36-37 genes as the standard gene number, theoretically there could be 36-37 intergenic regions. The existence of only 12 intergenic regions therefore indicates that there are no intergenic regions between 2/3 of adjacent genes. However, some lineages possess mitogenomes comprised mostly of noncoding DNA, with much larger numbers of intergenic spaces. For example, many nonbilaterian lineages possess large noncoding regions comprised of repetitive elements [6]. Expansion of noncoding regions has also been observed in isolated metazoan lineages, such as some nematodes, and even vertebrates [7–9]. Unfortunately, we could not identify a comprehensive review of the number of NCRs (nor overlapping genes) in metazoan mitogenomes, so many of these observations stem from our own studies of mitogenomes of invertebrates. Therefore, we added several references where suitable studies are available, and elsewhere modified the text in a way to avoid reliance on our personal observations: “As standard for mitogenomes of most animals [10], including isopods [11,12], the mitogenome of *A. magnifica* is also highly compact: we identified 12 non-coding regions (NCR; also referred to as intergenic spacers), only four of which were larger than 10 bp, and 14 gene overlaps. Although exceptionally large overlaps were identified in mitogenomes of some isopod lineages [13], there is no indication of such overlaps in *A. magnifica*. The largest (both 7bp) were found between *atp6/8* and *nad4/4L* genes, but the overlap between these genes is conserved in many metazoan lineages [14], including the isopods [12].”

In the last paragraph of this section, where we discuss rearrangement mechanisms, we added a few references that exemplify mitogenomes with highly expanded NCRs (in number and size). We also added a bit of discussion to explain better why this is unusual and important, and why we can use this to reject the TDLR mechanism: “The high compactness of isopod mitogenomes is very intriguing in the light of the rapid gene order evolution in this lineage. Often, taxa that exhibit high rates of mitochondrial architecture evolution also exhibit multiple expanded non-coding regions, sometimes spanning hundreds and even thousands of bases [7–9]. In some cases, these are believed to be a consequence of the tandem-duplication-random-loss (TDRL) rearrangement mechanism, wherein pseudogenes and non-coding DNA are generated and then again lost over the evolutionary time [9,10,15–17]. Therefore, high number of gene order rearrangements between closely related taxa and high compactness of their mitogenomes strongly suggest that TDRL mechanism is not the most parsimonious explanation for the high rate of gene order rearrangements in isopods. “

Page 13 Line 42-48: “As mitochondrial genes.....the third codon position (synonymous mutations)”. It's better to have the reference behind it.

R: We added a reference for each part of the sentence, and moved it a bit downwards in the paragraph, as it is a bit better logical fit: “As mitochondrial genes generally evolve under a strong purifying selection [18], nonadaptive hydrolytic deamination of bases generally has far more pronounced impact on the third codon position (synonymous mutations) of protein coding genes than on the first two codon positions [19].”

Reviewer: 2

Comments to the Author(s)

The manuscript is perfect and suitable for publication in your journal. But there are minor revision need to be done.

1. Page 4, Line 3-39; The known exception is not only one species, but two species, They are *Asellus aquaticus* (GC-Skew=-0.122) and *Janira maculosa* (GC-Skew=-0.026);

R: Initially we chose not to mention *Janira maculosa* as it is incomplete (<10,000 bp), but we can infer with some confidence that the entire Asellota clade exhibits negative skews, so we followed your advice and mentioned both species in the corrected manuscript.

2. Page 17, Line 59; Reference 63 is 2018; should use the newest literature (2019).

R: We updated the reference in the corrected version according to the instructions at the WORMS database (i.e., we refer specifically to the Isopoda web page).

References

1. Zhang D, Zou H, Hua C-J, Li W-X, Mahboob S, Al-Ghanim KA, Al-Misned F, Jakovlić I, Wang G-T. 2019 Mitochondrial Architecture Rearrangements Produce Asymmetrical Nonadaptive Mutational Pressures That Subvert the Phylogenetic Reconstruction in Isopoda. *Genome Biol. Evol.* **11**, 1797–1812. (doi:10.1093/gbe/evz121)
2. Wetzer R. 2002 Mitochondrial genes and isopod phylogeny (Peracarida: Isopoda). *J. Crustac. Biol.* **22**, 1–14. (doi:10.2307/1549602)
3. Wilson GDF. 2009 The phylogenetic position of the Isopoda in the Peracarida (Crustacea: Malacostraca). *Arthropod Syst. Phylogeny* **67**, 159–198.
4. Wolstenholme DR. 1992 Animal Mitochondrial DNA: Structure and Evolution. In *International Review of Cytology*, pp. 173–216. (doi:10.1016/S0074-7696(08)62066-5)
5. Bernt M, Braband A, Schierwater B, Stadler PF. 2013 Genetic aspects of mitochondrial genome evolution. *Mol. Phylogenet. Evol.* **69**, 328–338. (doi:10.1016/j.ympev.2012.10.020)
6. Lavrov DV, Pett W. 2016 Animal Mitochondrial DNA as We Do Not Know It: mt-Genome Organization and Evolution in Nonbilaterian Lineages. *Genome Biol. Evol.* **8**, 2896–2913. (doi:10.1093/gbe/evw195)
7. Zou H, Jakovlić I, Chen R, Zhang D, Zhang J, Li W-X, Wang G-T. 2017 The complete mitochondrial genome of parasitic nematode *Camallanus cotti*: extreme discontinuity in the rate of mitogenomic architecture evolution within the Chromadorea class. *BMC Genomics* **18**, 840. (doi:10.1186/s12864-017-4237-x)

8. Tang S, Hyman BC. 2007 Mitochondrial genome haplotype hypervariation within the isopod parasitic nematode *Thaumamermis cosgrovei*. *Genetics* **176**, 1139–1150. (doi:10.1534/genetics.106.069518)
9. Moritz C, Brown WM. 1987 Tandem duplications in animal mitochondrial DNAs: variation in incidence and gene content among lizards. *Proc. Natl. Acad. Sci. U. S. A.* **84**, 7183–7. (doi:10.1073/pnas.84.20.7183)
10. Boore JL. 1999 Animal mitochondrial genomes. *Nucleic Acids Res.* **27**, 1767–1780. (doi:10.1093/nar/27.8.1767)
11. Kilpert F, Podsiadlowski L. 2006 The complete mitochondrial genome of the common sea slater, *Ligia oceanica* (Crustacea, Isopoda) bears a novel gene order and unusual control region features. *BMC Genomics* **7**, 241. (doi:10.1186/1471-2164-7-241)
12. Zou H, Jakovlić I, Zhang D, Chen R, Mahboob S, Al-Ghanim KA, Al-Misned F, Li W, Wang G. 2018 The complete mitochondrial genome of *Cymothoa indica* has a highly rearranged gene order and clusters at the very base of the Isopoda clade. *PLOS ONE* **13**, e0203089. (doi:10.1371/journal.pone.0203089)
13. Doublet V, Ubrig E, Alioua A, Bouchon D, Marcade I, Marechal-Drouard L. 2015 Large gene overlaps and tRNA processing in the compact mitochondrial genome of the crustacean *Armadillidium vulgare*. *RNA Biol.* **12**, 1159–1168. (doi:10.1080/15476286.2015.1090078)
14. Taanman J-W. 1999 The mitochondrial genome: structure, transcription, translation and replication. *Biochim. Biophys. Acta BBA - Bioenerg.* **1410**, 103–123. (doi:10.1016/S0005-2728(98)00161-3)
15. San Mauro D, Gower DJ, Zardoya R, Wilkinson M. 2006 A hotspot of gene order rearrangement by tandem duplication and random loss in the vertebrate mitochondrial genome. *Mol. Biol. Evol.* **23**, 227–234. (doi:10.1093/molbev/msj025)
16. Fonseca MM, Harris DJ, Posada D. 2014 The Inversion of the Control Region in Three Mitogenomes Provides Further Evidence for an Asymmetric Model of Vertebrate mtDNA Replication. *PLOS ONE* **9**, e106654. (doi:10.1371/journal.pone.0106654)
17. Jühling F, Pütz J, Bernt M, Donath A, Middendorf M, Florentz C, Stadler PF. 2012 Improved systematic tRNA gene annotation allows new insights into the evolution of mitochondrial tRNA structures and into the mechanisms of mitochondrial genome rearrangements. *Nucleic Acids Res.* **40**, 2833–2845. (doi:10.1093/nar/gkr1131)
18. Nabholz B, Ellegren H, Wolf JBW. 2013 High Levels of Gene Expression Explain the Strong Evolutionary Constraint of Mitochondrial Protein-Coding Genes. *Mol. Biol. Evol.* **30**, 272–284. (doi:10.1093/molbev/mss238)
19. Min XJ, Hickey DA. 2007 DNA Barcodes Provide a Quick Preview of Mitochondrial Genome Composition. *PLOS ONE* **2**, e325. (doi:10.1371/journal.pone.0000325)